# Anatomy of the Palmar Region of the Carpus of the Dog

**DOI:** 10.3390/ani12121573

**Published:** 2022-06-18

**Authors:** Sonia González-Rellán, Andrés Barreiro, José Manuel Cifuentes, Patricia Fdz-de-Trocóniz

**Affiliations:** 1Department of Anatomy, Animal Production and Clinical Veterinary Science, University of Santiago de Compostela, 27002 Lugo, Spain; andres.barreiro@usc.es (A.B.); m.cifuentes@usc.es (J.M.C.); patricia.troconiz@usc.es (P.F.-d.-T.); 2Rof Codina Veterinary University Hospital, 27002 Lugo, Spain

**Keywords:** musculoskeletal, carpal canal, *flexor retinaculum*, *flexor carpi radialis*, median nerve, canine

## Abstract

**Simple Summary:**

Despite dogs being one of the main species in the veterinary field and having been used as animal models for human musculoskeletal research, the anatomical information about the palmar region of the canine carpus found in the literature is inconsistent. After the dissection and evaluation of complementary histology of 86 carpi obtained from 43 dogs, we provide precise information about the location, extensions, and anatomic relations of the palmar carpal components, with special reference to the carpal canal. The results of this study show that the anatomy of the palmar region of the canine carpus is more similar to human wrists than was previously thought. Numerous similar studies about the carpal anatomy of other species have been published; however, to our knowledge, this is the first article to clarify and describe reference information about the anatomy of the palmar region of the canine carpus.

**Abstract:**

The palmar region of the canine carpus is anatomically complex, and the information found in the literature about its anatomy is inconsistent. The aims of this prospective, descriptive, anatomic study were (1) the clarification and (2) the description of the precise anatomic composition of the palmar region of the canine carpus, with special reference to the *canalis carpi*. For this study, 92 cadaveric specimens were obtained from 46 dogs that had died for reasons unrelated to this study. Of these, 43 medium-to-large-breed dogs were randomly selected for the dissection of transverse slices of the carpus. Samples of the *flexor retinaculum* and *flexor carpi radialis* tendon and surrounding tissues were taken for complementary histology. For additional histology of the palmar structures in their anatomical position, three small breed dogs were randomly selected for obtaining transverse slices. The anatomic characteristics of the components of the palmar region of the canine carpus were qualitatively described, with special attention to the following structures: *flexor retinaculum*, *flexor carpi radialis* muscle, *arteria* and *vena mediana, nervus medianus*, *interflexorius* muscle, *flexor digitorum profundus* muscle, *canalis carpi*, and *arteria* and *nervus ulnaris*. The findings from this study provide reference information about the anatomy of the palmar region of the canine carpus.

## 1. Introduction

Musculoskeletal pain in the human upper limb constitutes a leading cause of disability in working populations [1], and wrist pain accounts for a high percentage [2]. Wrist pain may be the consequence of traumatic, inflammatory, infectious, neoplastic, metabolic, or degenerative conditions [3,4]. In addition, wrist pain can be caused by specific disorders of the palmar structures of the carpus, such as carpal tunnel syndrome, a clinical condition secondary to the compression of the *nervus medianus* leading to local numbness, weakness, and pain [5], that may develop independently or be associated and overshadowed by other diseases.

Dogs have been widely used as animal models in musculoskeletal research. However, based on our review of the literature, the use of this species to investigate specific palmar disorders is scarce.

To properly design, plan, and carry out any study on a particular animal model, a correct knowledge of its anatomy is mandatory. Despite the fact that the dog is one of the most studied species in the veterinary field, the information about the anatomy of its carpal region found in the literature is not consistent.

The palmar part of the canine carpus is an anatomically complex region consisting of the tendons of the flexor muscles of the carpus and digits (*flexor carpi radialis*, *flexor digitorum superficialis*, *flexor digitorum profundus*, and *flexor carpi ulnaris)*, the *flexor retinaculum*, the palmar carpal ligaments, and the neurovascular structures (mainly the *arteria* and *vena mediana*, the *nervus medianus*, and the *arteria* and *nervus ulnaris* (Figure 1). Barone [6] described the *canalis carpi* as the space located deep in the *flexor retinaculum*, containing the *nervus medianus*, the *arteria* and *vena mediana*, and the tendon of the *flexor digitorum profundus* muscle. The author stated that the tendon of the *flexor carpi radialis* muscle passes inside a synovial sheath through its own fibrous sheath on the medial side of the carpus, independent of the *canalis carpi*. Dyce et al. [7] did not provide a detailed description of the canine *canalis carpi*, but they explained that the *flexor digitorum profundus* muscle passes through it while the *flexor carpi radialis* and the *flexor digitorum superficialis* muscles remain apart from the canal. König and Liebich [8] defined the *canalis carpi* as the space delimited between the *flexor retinaculum* and the joint capsule of the carpus, containing the tendon of the *flexor digitorum profundus*, the *nervus medianus*, and the *nervus ulnaris.* On the other hand, Evans and de Lahunta [9] described that the *canalis carpi* has a wider extent, being the space generated deep to the *flexor retinaculum* and occupied by the tendons of the *flexor carpi radialis*, *flexor digitorum superficialis*, and *flexor digitorum profundus* muscles, the *nervus medianus*, the *arteria* and *vena mediana*, and the *arteria* and *nervus ulnaris*. This was also supported by Turan et al. [10,11,12] in later studies about the diagnosis of potential carpal canal syndrome in dogs. Ettema et al. [13], however, stated that the tendon of the *flexor digitorum superficialis* muscle is not located inside the *canalis carpi*.

The lack of agreement found in the veterinary literature impairs the understanding of the precise anatomical composition of this region in the canine carpus, complicating the comprehension of the real configuration of the *canalis carpi* and the anatomic relations of the carpal structures. Therefore, the objective of this study is the detailed description of the anatomy of the palmar region of the carpus of the dog, with special reference to the *canalis carpi*, in order to provide the basis for further studies of comparative anatomy and clinical research, that result in a better understanding of the carpal problems in both veterinary and human medicine

## 2. Materials and Methods

This study is a prospective, descriptive, anatomic design, part of a larger project for a doctoral thesis that was approved by the animal bioethics committee of the Rof Codina Veterinary University Hospital (HVU-RC) of the University of Santiago de Compostela.

The study’s methodology is based on similar procedures performed in other species such as humans [14] and equines [15]. For the anatomic identification of the carpal structures, the *Nomina Anatomica Veterinaria* and veterinary anatomic textbooks [6,7,8,9] were used as references.

For this study, 92 cadaveric specimens were obtained from 46 dogs that had died for reasons unrelated to this study. The specimens were acquired from the Service of Pathological Anatomy in the HVU-RC.

Of these, 43 medium-to-large-breed dogs were randomly selected for dissection according to inclusion criteria, which were weighing more than 20 kg, the absence of gross musculoskeletal disease in an orthopedic examination, having no history of musculoskeletal or nervous diseases in their medical records, and the lack of radiographic signs of musculoskeletal disorders in the carpal region. Of these 43 canine cadavers, 20 were females (46.5%) and 23 were males (53.5%). The mean age of the specimens was 6.73 years (range 1.5–12 years). Of the 43 carpi in group A, 21 were right and 22 were left. Of the 43 carpi in group B, 22 were right and 22 were left. The breeds of the dogs were Beagle (*n* = 5), mixed breed (*n* = 5), German Shepherd (*n* = 12), Golden Retriever (*n* = 3), Great Dane (*n* = 1), Labrador Retriever (*n* = 3), Mastiff (*n* = 8), Pointer (*n* = 2), Rottweiler (*n* = 1), Spanish Scenthound (*n* = 1), and St. Bernard (*n* = 2).

The 86 cadaveric specimens were transected at the level of the elbow and frozen at −18 °C until used. Limbs were thawed at room temperature for 8 h, and radiographs of the carpal region were performed to exclude signs of musculoskeletal disorders. Orthogonal views (dorsopalmar and mediolateral) were obtained. During the design of the protocol, the use of diagnostic techniques such as ultrasound, which is more sensitive to evaluate soft tissues than radiography, was considered. However, given that the normal ultrasonographic anatomy of the components of the palmar region of the canine carpus has not been published, in contrast with the dorsal structures [16], we decided not to use this technique to avoid potential bias. Both carpi of each dog were randomly assigned to group A or B, which included right and left specimens. The carpi of group A were dissected, and the carpi of group B were frozen again to −18 °C for 48 h in order to obtain cross-sections using a hand saw.

In the specimens in group A, the hair was clipped from the middle antebrachial region to the distal metacarpus. The skin was longitudinally incised at the dorsolateral aspect, then reflected, and the overlying fascia, connective tissue, and fat were removed. The dissection was performed sequentially, in a medial to lateral and superficial to deep fashion. The studied region of the included soft tissue structures was the area between the distal part of the antebrachium and the proximal aspect of the metacarpus (Figure 1). The anatomic characteristics (location, shape, size, and topographic relations) of the structures of the palmar region of the canine carpus were qualitatively described and documented, with special attention to the following structures: *flexor retinaculum*, *flexor carpi radialis* muscle, *nervus medianus, arteria* and *vena mediana*, *interflexorius* muscle, *flexor digitorum profundus* muscle, *canalis carpi* and *arteria*, and *nervus ulnaris*. In group B, 3 mm thick slices were cut with a hand saw, and photographs of both faces were taken and correlated with the findings of the dissection. Complementary histology was performed in samples of the *flexor retinaculum* and *flexor carpi radialis* tendon and surrounding tissues.

For complementary histology of the palmar structures in their anatomical position, three small breed dogs were randomly selected for obtaining transverse slices. The inclusion criteria were the same as for the other dogs, except for the weight. Therefore, three Yorkshire specimens were included, one male and two females, with a mean age of 12.03 years (range 10.25–13.83 years). The carpi were transected (3 mm thick slices), and the samples were fixed by immersion in 10% buffered formaldehyde for a minimum of 48 h. After decalcification in Shandon TDB-1TM, given the difficulty of cutting the calcified tissue, the samples were embedded in paraffin according to standard laboratory procedures, and sections of 5 μm thickness were mounted onto silanized slides and dried overnight at 37 °C. The sections were stained with H and E for routine histological analyses, Azan trichrome stain, OMSB (combined orcein and Martius scarlet blue) for elastic fibers, and Picro Sirius Red for collagen fibers. Samples were evaluated in terms of the general structure and anatomical relations with an Olympus AX70 microscope and a coupled digital camera Olympus DP74.

## 3. Results

The anatomy of the palmar region of the carpus was consistent in all canine carpal specimens examined.

### 3.1. Flexor Retinaculum

Once the skin and subcutaneous fat were removed (Figure 2A), an oblique transverse thickening of the fascia was identified, extending from the dorsal carpal and metacarpal fascia to the palmar fat pad (Figure 2B). It was composed of two transverse fibrillar thickenings of the fascia connected by oblique and thin fibrous bands and constituted the superficial part of the *flexor retinaculum* (Figure 2C). The proximal one extended from the fascia of the dorsal carpal region, in continuity with the *extensor retinaculum*, to the peritendineum of the *flexor digitorum superficialis* tendon, to which it was fused. The distal part extended from the sesamoid bone of the abductor pollicis longus muscle to the subcutaneous tissue of the palmar pad. The proximal thickening covered the tendon of the *flexor carpi radialis* muscle, while the distal one passed superficially to the *flexor digitorum superficialis* tendon. The distal end of the antebrachial fascia was obliquely directed from the palmar aspect of the carpus towards the dorsomedial region of the metacarpus, forming a flat band that resembled an aponeurosis. The superficial part of the *flexor retinaculum* was located superficially to this distal aponeurotic-like band of the antebrachial fascia. However, they were partially fused. The direction of the fibers allowed a clear differentiation of these structures, being transverse to those of the *retinaculum* and longitudinal and oblique to those of the antebrachial fascia.

After making a central incision in these fascial thickenings, the dissection continued with their removal (Figure 2D), together with the tendon of the *flexor digitorum superficialis* muscle. The described proximal band of the superficial part of the *flexor retinaculum* was fused at the medial region of this tendon, both to the antebrachial fascia and to a deep, strong, and well-defined transverse fibrous band (Figure 2E) that extended from the accessory carpal bone to the styloid process of the radius, the intermedioradial bone, and the first carpal bone. This transverse fibrous band was the deep part of the *flexor retinaculum* and constituted the roof of the *canalis carpi*. It sent a transverse extension proximally and medially towards the radius, which was much thinner, and covered the tendon of the *flexor carpi radialis* muscle and the adjacent *nervus medianus* and *arteria* and *vena mediana*.

Both parts of the *flexor retinaculum* consisted of dense connective tissue with collagen fibers predominantly aligned perpendicularly to the adjacent tendons and surrounded by loose connective tissue and fat.

### 3.2. Canalis Carpi

The *canalis carpi* was seen in transverse sections as an ovoid and well-delineated non-distensible space (Figure 3A,B), extending between the antebrachiocarpal and the carpometacarpal joints. With the carpus in dorsal recumbency, the deep part of the *flexor retinaculum* acted as the roof of a space created between it and the common palmar ligament, which constituted the floor (Figure 2E,H). The medial wall was formed by the medial fixations of the fused antebrachial fascia to the deep part of the *flexor retinaculum* and the common palmar ligament. This medial wall separated the structures that passed inside the canal from the tendon of the *flexor carpi radialis* muscle. A thin fibrous septum that extended vertically from the lateral region of the deep part of the *flexor retinaculum* to the palmar common ligament constituted the lateral wall and isolated the structures of the *canalis carpi* from the *flexor carpi ulnaris* muscle, the *nervus ulnaris*, and the accessory bone and its ligaments. Histologically, this septum appeared as a thin band of dense collagen fibers. The *canalis carpi* were occupied by the tendons of the *flexor digitorum profundus* and the *interflexorius* muscles, as well as the *arteria* and *vena mediana* and the *nervus medianus* (Figure 2F,G and Figure 3A,B).

The compound tendon of the *flexor digitorum profundus* muscle consisted of the fusion of the tendons of the three muscular bellies located at the deep part of the caudal forearm. Proximally to the antebrachiocarpal joint, these tendons fused and formed a flattened concave merged tendon that entered the *canalis carpi* and occupied its deepest part. Distally to the canal, it split and sent a branch to each digit. Along the canal, a paratenon was identified as a peritendinous sheet composed of loose fibrillar tissue and fat. Palmar to the concavity of the merged tendon, the space was filled by the rest of the structures of the canal.

The *nervus medianus* and the *arteria* and *vena mediana* ran adjacent to the *flexor carpi radialis* tendon in the palmaromedial region of the antebrachium and deviated laterally to enter the *canalis carpi* proximal to the carpus. The artery was central, the vein was lateral, and the nerve was medial. Lateral to these neurovascular components, the two thin tendons of the *interflexorius* muscle passed towards the metacarpus. All these structures were interconnected and linked to the paratenon of the *flexor digitorum profundus* muscle by loose connective tissue and a small amount of fat (Figure 2F).

### 3.3. Flexor Carpi Radialis Muscle

After the dissection of the *canalis carpi* and the complete removal of its components, the common palmar ligament appeared as a smooth surface (Figure 2H). The medially located tendon of the *flexor carpi radialis* muscle was then examined (Figure 4A,B).

The muscular belly of the *flexor carpi radialis* muscle laid superficially and proximally in the palmaromedial region of the antebrachium, running its tendon medially and deeply towards the carpal bones, where it entered inside an independent tunnel. The common palmar ligament fused medially to the deep part of the *flexor retinaculum* and the antebrachial fascia, adjacent to their attachments to the medial carpal bones. These medially fused structures covered the tendon of the *flexor carpi radialis* muscle, complementing superficially its tunnel, which was well recognized during macroscopical examination. In transverse sections, the tunnel was seen as a U-shaped band of dense collagen fibers that surrounded palmarly the tendon from approximately 1 cm proximal to the antebrachiocarpal joint to its insertion at the basis of the second metacarpal bone. However, the deepest part differed along the carpus. The proximal segment of the tunnel located between the distal region of the radius and the antebrachiocarpal joint was seen as a thick fibrous structure (Figure 4B) that consisted of a U-shaped band of dense collagen fibers intercalated by cells morphologically compatible with fibroblasts and a small number of chondrocytes (Figure 4C). At its deepest part, the collagen bundles were partially replaced by a fusiform area of fat and loose connective tissue that contained blood vessels. At the intermedioradial bone, the tendon passed through a sulcus. The U-shaped band of collagen fibers of the tunnel was thinner, and it was fused dorsally with a fibrocartilaginous area associated with the sulcus (Figure 4D). At the mid-carpal joint, the tendon was in contact with the articular capsule covering directly an adjacent intra-articular palmar mid-carpal ligament attached to the second carpal bone. At the carpometacarpal joint, the tendon was intra-articular and in direct contact with the synovial fluid. Distally and up to the enthesis, the tunnel was flattened and thickened palmarly, while absent at the deepest region (Figure 4E). The tunnel was composed of dense connective tissue and numerous cells morphologically compatible with chondrocytes.

Proximally, the tendon partially occupied the space of the tunnel. However, distally it filled most of the lumen. At the enthesis, the size of the tendon was greater.

### 3.4. Nervus Ulnaris

The *nervus ulnaris* was located at the lateral aspect of the palmar region of the carpus. It ran medially and deep to the *flexor carpi ulnaris* muscle and superficially to the *flexor digitorum profundus* muscle (Figure 2G and Figure 5A,B). Proximally to the accessory carpal bone, it sent the superficial branch. The deep branch continued to the medial part of the accessory carpal bone together with the *arteria ulnaris*, both separated from the *canalis carpi* by a thin fibrous septum that constituted its lateral wall. Then, at the level of the IV carpal bone, the nerve and artery entered a space created by the medial accessory-metacarpal ligament and the accessory carpal bone, where they were surrounded by adipose tissue. The nerve emerged distally between the *abductor* and *flexor digiti V* muscles and split to send branches to the digits.

## 4. Discussion

The anatomy of the main structures of the palmar region of the carpus of the dog is described in this study.

We found a strong similarity between the canine and human *flexor retinaculum*, given that both are composed of a superficial part, a specialized thickening of the fascia, and a deep one, a wider fibrous band, resembling a ligament. In humans, the superficial part is also constituted by proximal and distal sections [17], as we have seen in the dog. The superficial part showed the characteristics of a typical *retinaculum*, being a specialized thickening of the fascia that covered flexor tendons. However, the deep part was a strong fibrous band that resembled more a ligament and, despite being a deep continuation of the superficial part, it additionally showed attachment to bones. In human wrists, this deep structure has been named the “transverse carpal ligament” [18], but the use of this term is controversial in the literature. Despite its anatomic resemblance to a ligament, its main function appears to be the reinforcement of the underlying tendons, which corresponds to that of a *retinaculum*. Stecco et al. [19] reviewed the terminology of these structures in humans, concluding that naming both parts as the “*flexor retinaculum*” may induce confusion about which part is alluded to, so the term “transverse carpal ligament” should be used to specifically refer to the deep part that acts as the roof of the carpal canal (considering the carpus in dorsal recumbency).

In the veterinary literature, there is no specific terminology for this deep part. As previously explained, the *flexor retinaculum* in dogs is briefly mentioned among the main anatomic reference sources and in the NAV, and a detailed description is lacking. Nevertheless, Nickel et al. [20] state that the *flexor retinaculum* corresponds to the transverse carpal ligament that forms the *canalis carpi.* Therefore, considering our results, we propose to term this structure in the canine carpus as the “deep part of the *flexor retinaculum*”, understanding that it is more accurate given its function and location.

In this study, the *canalis carpi* in dogs was described as a non-distensible and well-defined oval space delimitated palmarly by the deep part of the *flexor retinaculum*, laterally by a thin fibrous septum, dorsally by the common palmar ligament that covers the palmar region of the carpal bones, and medially by the fixations of the fused antebrachial fascia to the deep part of the *flexor retinaculum* and the common palmar ligament. It contained the tendons of the *flexor digitorum profundus* and the *interflexorius* muscle, as well as the *nervus medianus* and the *arteria* and *vena mediana*. The inconsistency found in the literature is probably related to the non-differentiation between the superficial and deep parts of the *flexor retinaculum*. Dyce et al. [7], Evans and de Lahunta [9], and Turan et al. [10,11,12] made their descriptions assuming that the *flexor retinaculum* was only constituted by the superficial part, which led to the inclusion of other tendons and neurovascular structures inside the hypothetical *canalis carpi*. However, the superficial part of the *retinaculum* did not generate a true canal below because, by definition, a canal is a tubular duct or channel. The presence of the superficial part of the *retinaculum* did not create a canal itself, and there is no defined canal beneath the *extensor retinaculum* of the carpus. Barone [6] and Ettema et al. [13] did describe the *canalis carpi* only as the space underlying the deep part of the *flexor retinaculum*. Our results support this and correlate with human anatomy, where the *canalis carpi* is a space in the palmar region of the carpus formed between the deep part of the *flexor retinaculum*—the transverse carpal ligament—and the carpal bones, which contains the tendons of the *flexor digitorum superficialis*, *flexor digitorum profundus*, and *flexor pollicis longus* muscles, as well as the *nervus medianus* [21]. In humans, the entrapment of this nerve originates a well-known clinical condition, carpal tunnel syndrome, which is the most common disorder of the hand [19] and the most common peripheral mononeuropathy [22]. In veterinary medicine, to the best of our knowledge, the clinical significance of this condition in dogs has not been established. However, its potential occurrence has been proposed [11].

In this study, the tendon of the *flexor carpi radialis* muscle was described as passing through its own fibrocartilaginous tunnel, independent of the *canalis carpi*. Dyce et al. [7] and Evans and de Lahunta [9] included the course of this tendon inside the *canalis carpi*, not mentioning this specific tunnel. König and Liebich [8] and Nickel et al. [20] described this tendon outside the *canalis carpi*, passing into a synovial and tendinous sheath, respectively. Our results support the description of Barone [6] and also correlate with human anatomy. In the radial side of the palmar region of the human wrist, the tendon of the *flexor carpi radialis* muscle passes through its own osteofibrous tunnel independent of the *canalis carpi* [23], to insert in the base of the second, and often third, metacarpal bones [24]. In humans, this tunnel also does not completely surround the tendon through its entire course; therefore, the *flexor carpi radialis* muscle lies in close contact with the shared joint capsule of the scaphotrapeziotrapezoid (STT) or triscaphe articulation, which may lead to the extension of disease from one compartment to the other [25]. Additionally, the anatomic characteristics of the tendon and the non-distensible nature of the tunnel are related to tendinitis [23] and stenotic tendinopathy [26], also known as the *flexor carpi radialis* tunnel syndrome [27], clinical conditions characterized by inflammation and pain. To our knowledge, there is no published information about these injuries in dogs.

Evans and de Lahunta [9] and König and Liebich [8] described the path of the *nervus ulnaris* inside the *canalis carpi*. Our results, however, place the *nervus ulnaris* lateral to it, separated by a septum of connective tissue. These findings also correlate with human anatomy. In the ulnar side of the palmar region of the wrist, the *nervus ulnaris* also passes outside the *canalis carpi*, running through a tunnel termed the Guyon’s canal [28,29]. This is clinically important given that the nerve may suffer entrapment, which leads to a condition known as the ulnar tunnel syndrome; though less prevalent than the carpal tunnel syndrome, it constitutes the fourth most common entrapment neuropathy in the literature [30], and the second most frequent in the human upper extremity after carpal tunnel syndrome [31]. In the canine carpus, the nerve passes between the accessory carpal bone and the lateral septum of the *canalis carpi*. Then, it continues deep to the medial accessory-metacarpal ligament, where it is surrounded by fat tissue. To the best of our knowledge, there is no published information about the entrapment of this nerve in canines.

Dogs have been widely selected as animal models in musculoskeletal research and have been used in multiple studies about osteoarthritis, both naturally and induced-occurring [32,33]. This species has been considered the closest to a gold standard as an experimental model for osteoarthritis because of its similarities with humans in anatomy, disease progression, and outcome [32]. Furthermore, the size of these animals has shown other advantages as an experimental model in comparison with smaller species such as mice and rabbits, such as the possibility of using antemortem diagnostic and monitoring assessments (i.e., diagnostic imaging techniques). However, given that dogs are quadrupeds, their biomechanics differ from humans; in the resting position, the flexor compartment of the carpus is tensioned, which is not the case in bipedal subjects. This concept should be considered when canines are selected as animal models in musculoskeletal research. Although they have been used in studies centered on the carpus [34,35,36] and the flexor tendons [13], based on our review of the literature, the use of this animal to investigate the above-described specific palmar disorders is scarce.

The confusing anatomic descriptions published so far in the veterinary literature may have prevented the proper application and interpretation of diagnostic techniques. Further research is needed about all these carpal conditions in dogs in veterinary medicine, and the use of this species as animal models to investigate the aforementioned diseases in human wrists merits promotion.

The limitations of the present study are, first, the heterogeneity of the included dogs, which did not permit obtaining measurements of the structures transferable to the whole canine population. Additional studies may be interested in providing these data on particular breeds. Second, our results were mainly obtained from a gross anatomy approach, presenting a complete macroscopic perspective of the location, extensions, and anatomic relations of the main structures of the palmar side of the canine carpus. However, more histologically specific studies may be needed to determine the precise microscopic composition of these elements. Third, radiography is a sensitive technique to evaluate bones, but its sensitivity for the examination of soft tissue is poor. The use of more sensitive techniques such as ultrasound to rule out disease in these structures would have been optimal during the selection process of the cadaveric specimens. However, despite the fact that the normal ultrasonographic anatomy of the dorsal region of the carpus of the dog has recently been published [16], the same information about the palmar structures is lacking. For that reason, this technique has not been included in the protocol of this study, but we consider that once a correct description of the anatomy of this region is published, further studies about the normal and pathological ultrasonographic appearance of the palmar structures of the canine carpus will be highly interesting.

## 5. Conclusions

The findings of this study provide an anatomic description of the palmar side of the canine carpus and show that the anatomy of the carpi of dogs and humans is more similar than was previously considered. These findings may constitute a basis in the research of musculoskeletal injuries, both to improve the use of canine models for human diseases and to expand the information about canine carpal diseases inside the veterinary field.

## Figures and Tables

**Figure 1 animals-12-01573-f001:**
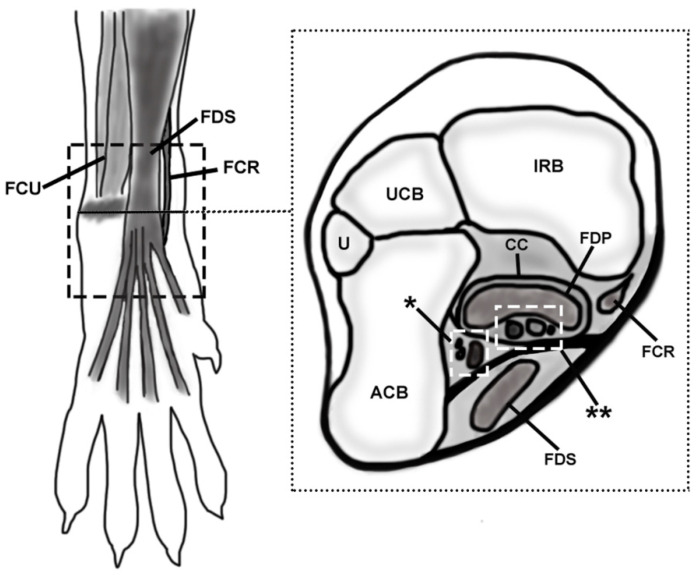
Schematic representation of the simplified anatomic location of the palmar structures of the canine carpus. The discontinuous lined rectangle in the left image (a palmar view of the carpus) delimitates the studied area. A transverse slice was drawn to show the deeper structures at the level of the accessory carpal bone (represented with a shadow in the left image and as ACB in the right one). The level of the transverse slice is signaled with a black line in the palmar view. FCU, *flexor carpi ulnaris*; FDS, *flexor digitorum superficialis*; FCR, *flexor carpi radialis;* U, ulna; UCB, ulnar carpal bone; IRB, intermedioradial bone; CC, *canalis carpi*; FDP, *flexor digitorum profundus*; asterisk, ulnar neurovascular package (*arteria, vena*, and *nervus ulnaris*); double asterisk, median neurovascular package (*arteria* and *vena mediana, nervus medianus*) and *interflexorius* muscle.

**Figure 2 animals-12-01573-f002:**
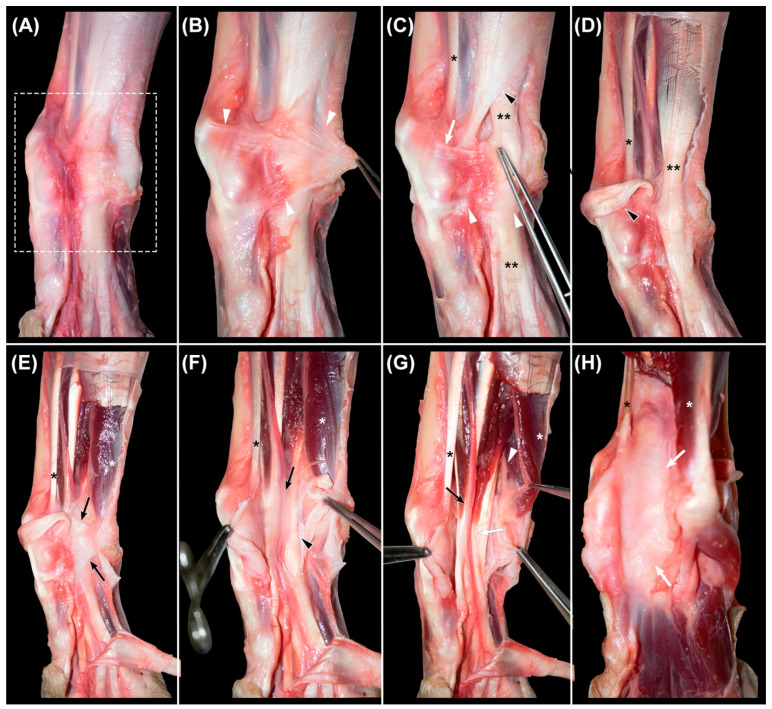
Sequential images of the dissection of the palmar region of the canine carpus. (**A**): area under study (rectangle) after removing the skin and subcutaneous fat. (**B**): identification of the superficial part of the *flexor retinaculum* (arrowheads). (**C**): dissected proximal (white arrow) and distal (white arrowheads) bands of the superficial part of the *flexor retinaculum* and the distal attachment of the fascia of the antebrachium (black arrowhead), and (**D**): aforementioned structures reclined. Double asterisk (**C**,**D**): tendon of the flexor digitorum communis muscle. (**E**): identification of the deep part of the *flexor retinaculum* (black arrows) and (**F**): its incision to access the *canalis carpi*. The neurovascular structures and tendons of the *interflexorius* muscle surrounded by loose connective tissue and a small amount of fat (black arrow), overlying the tendon of the *flexor digitorum profundus* muscle (black arrowhead). (**G**): dissection of the *arteria* and *vena mediana* and *nervus medianus* (black arrow) and identification of the tendons of the *interflexorius* muscle (white arrow). The *nervus ulnaris* (white arrowhead), lateral to the *canalis carpi* and deep to the *flexor carpi ulnaris* muscle (white asterisk). (**H**): after the removal of the structures of the *canalis carpi*, the underlying palmar common ligament is seen as a smooth surface (white arrows). Black asterisk (**C**–**H**): tendon of the *flexor carpi radialis* muscle.

**Figure 3 animals-12-01573-f003:**
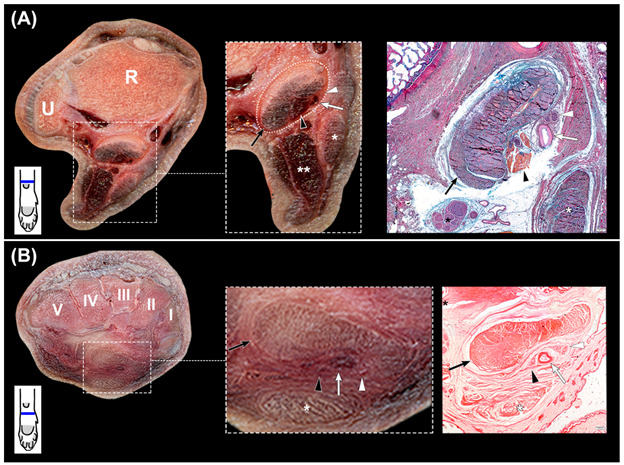
(**A**,**B**): transverse slices of the canine carpus. (**A**): at the distal part of the radius (R) and ulna (U), with magnification of the *canalis carpi* and complementary histologic section stained with OMSB. (**B**): at the level of the carpal bones (I–V) with magnification of the *canalis carpi* and complementary histologic section stained with Picro Sirius Red. The *canalis carpi* are outlined with dots (rectangle), and its components are shown in the magnified image and in the stained sections. The Picro Sirius Red stained section shows the collagen fibers that separate the structures of the *canalis carpi* from the *nervus ulnaris* (black asterisk) and the *flexor digitorum superficialis* muscle (white asterisk). Black arrows: *flexor digitorum profundus* muscle; black arrowheads: *interflexorius* muscle; white arrows: *arteria mediana*; white arrowheads: *nervus medianus*; double white asterisk: *flexor carpi ulnaris* muscle. In the left corner below the transverse slices, a scheme represents their corresponding level at the carpus.

**Figure 4 animals-12-01573-f004:**
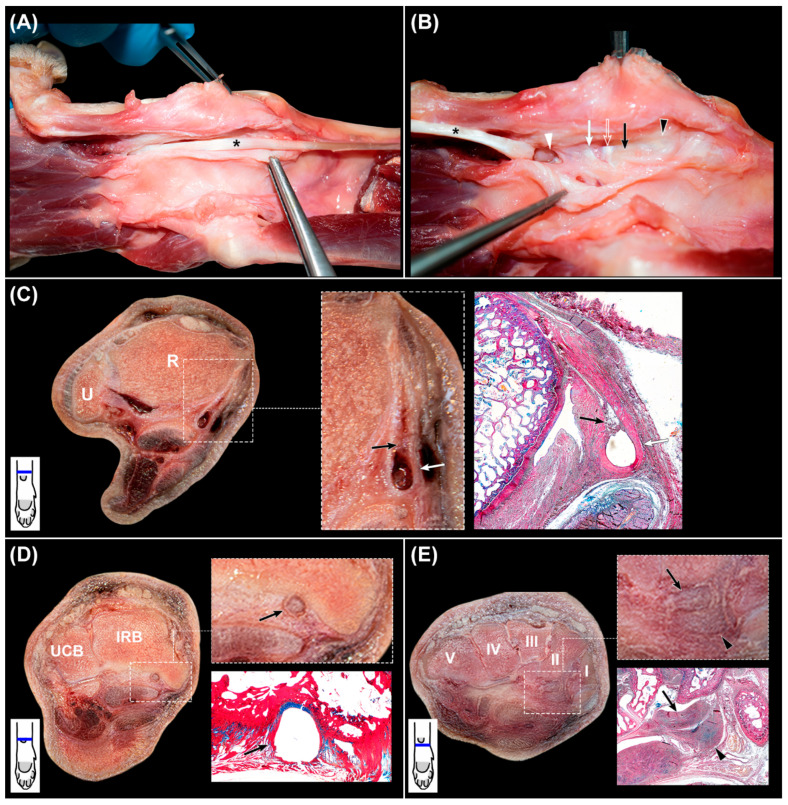
(**A**,**B**), dissection of the *flexor carpi radialis* tendon (asterisk) and its tunnel. In both images, proximal region is at the right. In (**B**), the tendon has been removed to expose the deep aspect of the tunnel. Black arrowhead: fibrous part at the level of the antebrachiocarpal joint; black arrow: sulcus of the intermedioradial bone; open arrow: articular capsule of the mid-carpal joint; white arrow: mid-carpal ligament; white arrowhead: carpometacarpal joint. (**C**), transverse slice of the canine carpus at the distal part of the radius (R) and ulna (U) with magnification of the *flexor carpi radialis* tendon tunnel (white arrows) and corresponding histologic section stained with OMSB. Black arrows: area of fat, loose connective tissue, and blood vessels. (**D**): transverse slice of the canine carpus at the intermedioradial bone (IRB) and ulnar carpal bone (UCB) with magnification of the *flexor carpi radialis* tendon tunnel (black arrows) and corresponding histologic section stained with OMSB. (**E**): transverse slice of the canine carpus at the carpal bones (I–V) with magnification of the *flexor carpi radialis* tendon tunnel (black arrowheads) and corresponding histologic section stained with OMSB. Black arrows: *flexor carpi radialis* tendon. In the left corner below the transverse slices, a scheme represents their corresponding level at the carpus.

**Figure 5 animals-12-01573-f005:**
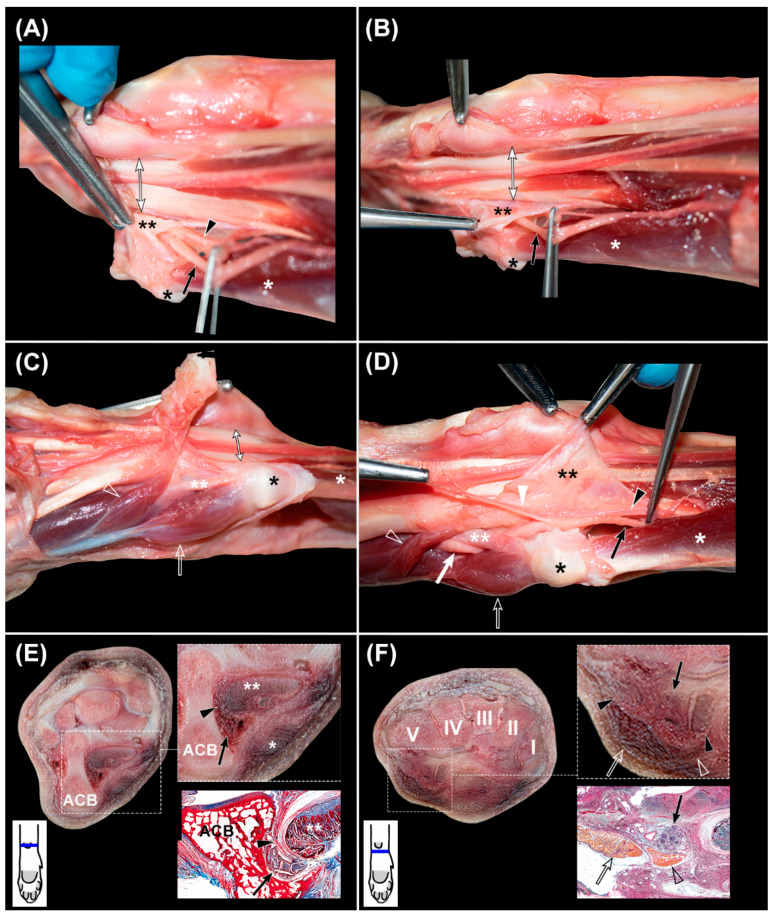
(**A**–**D**), sequential dissection of the *nervus ulnaris*, whit the distal region of the carpus located at the left. (**A**,**B**) are slightly oblique. Double black asterisk: lateral septum of the *canalis carpi*; black asterisk: fat pad superficial to the accessory carpal bone; white asterisk: *flexor carpi ulnaris* muscle; black arrowhead: *arteria ulnaris*; black arrow: *nervus ulnaris* proximal to the accessory carpal bone; double-headed arrow: *canalis carpi*; double white asterisk: accessory-metacarpal ligament; white arrowhead: superficial branch of the *arteria* and *nervus ulnaris*; white arrow: *nervus ulnaris* distal to the accessory-metacarpal ligament; open arrow: *abductor digiti V* muscle; open arrowhead: *flexor digiti V* muscle. (**E**,**F**): complementary transverse slices of the carpus at the level of (**E**), the accessory carpal bone (ACB) and (**F**), the carpal bones (I–V), with magnification of the *nervus ulnaris* and supporting histologic sections stained with OMSB. In the left corner below the transverse slices, a scheme represents their corresponding level at the carpus.

## Data Availability

Data available on request due to restrictions eg privacy or ethical.

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
