# Peer review of "Anatomy of the Palmar Region of the Carpus of the Dog"

_animals, 2022, doi:10.3390/ani12121573_

Round 1
Reviewer 1 Report
The paper deals with a key anatomical structure of the thoracic limb from the point of view of biomechanics. However, the detailed structure of the carpal tunnel in relation to the structures passing through it has not been described in detail in the dog. The research material is sufficient. The authors should redraft the text to some extent. The sentence (line 5) should be completed. They should write the model of the microscope used in Material and Methods. Moreover, they should use Latin nomenclature if they mention a certain structure for the first time, especially since they do so in reference to muscles. As the authors refer to the knowledge available in academic textbooks, rightly pointing out their generality, they should include the following items in the discussion:
1.Nickel R., Schummer, A., Seiferle E.: Lehrbuch derAnatomie der Haustiere. Band I. Bewegungsapparat.Parey Verlag 2004
2. Konig H., E., Liebich H-G.: Anatomie der Haussaugetiere. Schattauer 2012- English version can be used.
Detailed notes are included in the pdf. ofpublication

Reviewer 2 Report
The study has used a very significant number of specimens and presents some brilliant images of dissections. However, I think a series of changes are necessary to recommend it for publication:
The introduction is too simple, it does not mention important aspects that should be explained as a preamble to the study.
Because several comparisons are made with man and the dog is offered as an experimental model, explain the differences between the carpus of man and dog (bones that make up each of the rows and their arrangement, some drawings could give greater clarity) . It should be clear why it can be a model despite the anatomical (eg psiform bone) and mobility differences in both species.
Describe carpal tunnel syndrome in addition to citing it (line 42).
There are few bibliographic citations despite affirming that this region has been the focus of many studies. For example, add a bibliographic citation on line 44 to support what is stated.
Define the carpus in the introduction and differentiate the concepts of canal and carpal tunnel (which are not synonyms). The term tunnel is more appropriate, it should be used when describing this structure without being open.
In the introduction or the results, put a schematic or cross-sectional drawing of the carpal region showing the different compartments with their contents to make it easier for non-expert readers to follow the description.
When the cross sections are presented, put a diagram of what level they have been made. It is a wide region and the level of the presented cut must be indicated.
Reviewer 3 Report
Anatomy of the palmar region of the carpus in the dog.
The paper accurately and in-depth describes the study region using the dissection and cross-sectioning method.
The number of samples used appears adequate in order to obtain reliable results.
The images attached to the study are very explanatory and make the reading of the text easy and clear.
Some remarks:
47: Can the anatomical region of the dog be used as a study model for humans? The anatomy is comparable, the biomechanics highly different. In the quadruped, the forelimb, resting on the ground, keeps the flexor compartment almost constantly under tension, unlike bipedal subjects. I believe that at least one reference to these concepts must be made
84: orthopedic examination; this evaluation includes dynamic (walking) and static functional tests as well as manipulation carried out on the alive and conscious patient. Considering that these are dead dogs, how long before death was the orthopedic examination performed? I believe it is important to indicate this value to understand the reliability of the orthopedic examination for the purpose of excluding a pathology.
85: "radiographic signs of musculoskeletal disorders": radiographic examination has high sensitivity for skeletal diseases but low sensitivity for detecting most muscle diseases. This study analyzes a region consisting mainly of tendons, retinacles, ligaments. Don't the authors believe that the radiographic study may not allow to totally exclude the affections of the aforementioned structures? Probably more sensitive diagnostic means (ultrasound) would not give more adequate status? I believe the concept should be clarified in the text in the materials and methods section and, perhaps, listed among the study limits.
366-368: this statement refers to a scientific work (28). The McCoy paper (2015) speaks exclusively of osteoarthritis, not of "musculoskeletal research" in general.
There is no correlation between osteoarthritis and carpal tunnel syndrome except for rheumatoid arthritis in humans.
Therefore, the cited paper does not support the statement that it should be rewritten.
Reviewer 4 Report
The manuscript describes the topography of tendons and other connective structures in the palmar site of the carpus of dogs. The Authors point out the importance of this study as model for human utility, whereas they could touch on this aspect by anthropocentric point of view. This work is important firstly for dogs and for veterinary clinicians and then for humans. The figures are of good quality. The methods are adequate. It would have been interesting to compare with more small dogs and, overall, between dogs involved in works (hunters, sheepdogs, etc) and dogs living in house (pets). In the materials it is not well clear how many dogs have been used (46 or 43 ?) and the origin of these dogs.
In the pag 2, line 52 : "the c" what means?
Round 2
Reviewer 1 Report
The current version of the manuscript can be published. The authors described the carpal tunnel and associated structures in detail, which they related to the available literature.
Author Response
Thank you once again for taking the time to further review our manuscript and responses.
Reviewer 2 Report
The authors have done a great job improving the manuscript. Above all, the added drawings help to better interpret the results of the study. I recommend this manuscript for publication in its present form. Congratulations.
Author Response
Thank you for taking the time for reviewing the new draft and responses. Your previous comments were constructive and made us to reflect about the manuscript and improving it.